# The effect of photoperiod, environmental temperature and wind speed on external quality of free-range turkey eggs

José Ignacio Salgado Pardo[1], Juan Vicente Delgado Bermejo[1],
Antonio González Ariza[2]*, José Manuel León Jurado[2],
María Esperanza Camacho Vallejo[3]

1 Department of Genetics, Faculty of Veterinary Sciences, University of Córdoba, Córdoba, Spain,
2 Agropecuary Provincial Center of Córdoba, Provincial Council of Córdoba, Córdoba, Spain,
3 Department of Agriculture and Ecological Husbandry, Area of Agriculture and Environment, Andalusian Institute of Agricultural and Fisheries Research and Training (IFAPA), Alameda del Obispo, Córdoba, Spain

* angoarvet@outlook.es

## Abstract

Due to the lack of research in free-range poultry, the influence of environmental factors on laying remains underdeveloped nowadays. Therefore, the present study aimed to determine the influence of meteorological events and the moon cycle on the weight and shape index of turkey eggs produced in alternative systems. To this aim, 194 eggs laid by Andalusian turkey hens raised outdoors were collected and daily measured for 14 months. Seven categories were obtained attending to their weight (heavy, medium, and light) and shape index (sharp, standard, and round), and performed as dependent variables in the discriminant canonical analysis (DCA). As explanatory variables, meteorological and moon cycle features were collected from repositories consulting the day before each egg was laid. Minimum pressure, moon phase, and maximum temperature reported multicollinearity problems (VIF > 5.0) and were removed from further analysis. Moderate correlations among climatic variables influencing egg quality were found and ranged from + 0.574 to − 0.537, involving maximum gust speed in both cases. Variables exhibiting explanatory power in the DCA were sunshine hours ($\lambda = 0.691$, $F = 13.950$), minimum temperature ($\lambda = 0.874$, $F = 4.476$), and maximum gust speed ($\lambda = 0.944$, $F = 1.841$). Results evidence alternative productions present a different environmental exposure than indoors, as well as highlight the suitability of this breed for free range systems.

## Introduction

Alternative poultry production has arisen in developed countries during the last decades [1,2] due to the market's demands for natural products obtained from sustainable and animal welfare-friendly systems [3]. In this context, traditional farming as backyard and extensive regimes are being restored [4]. In addition to the food

**Data availability statement:** All relevant data are within the manuscript and its Supporting information files.

**Funding:** This research was funded by FEDER project P20_00893 and, during the covering period of a pre-doctoral contract (FPU Fellowship, with code FPU21/00010), the Spanish Ministry of Science and Innovation.

**Competing interests:** The authors have declared that no competing interests exist.

contribution, alternative farming has a positive impact on overall biodiversity [5], the preservation of local breeds, and the fixation of rural populations [1]. This situation contrasts dramatically with that of developing countries, where backyard production is not a preference, but a necessity. Depending on the region, backyard poultry comprises about 30–80% of the total census [6] and can contribute up to 90% of poultry production [7] in addition to its social and cultural implications [8].

In this context, turkey is one of the most widespread poultry species in the world [9] due to its high heat tolerance and grazing aptitude [8]. Little attention has been paid to turkey eggs as food, possibly due to their low natural suitability for laying [10]. Nevertheless, backyard turkey eggs are an important source of animal protein in developing countries [8]. On the other hand, in developed countries, turkey eggs are mainly allocated to hatching [11], and consumption is concentrated in rural areas, based on local breeds reared in backyards [12]. However, thanks to the particularities of the species [13], organic turkey eggs from local breeds are considered niche products with high potential in haute cuisine [14].

Outdoor systems face high environmental exposure [15], which accounts for up to 30% of the overall variance in external egg traits [16]. Weather is one of the main environmental factors influencing turkey performance, particularly temperature, humidity, and photoperiod [17]. High-yielding hybrid strains struggle in uncontrolled environments, leading to losses in productivity and welfare issues [2,3], while indigenous, locally adapted genotypes are more suitable for outdoor systems [18]. Few adapted turkey landraces have been described in the Mediterranean basin, where extreme climatic conditions are reached [19]. To this respect, the Andalusian turkey is a light and rustic population from the south of Spain commonly reared in backyards and pasture regimes [20]. Although this genotype could be direct descendants of the first turkeys to arrive in Europe in the 16th century [12,21], they are currently in a severely endangered status. In this context, the promotion of the consumption of close by, organic eggs to encourage the use of the breed and improve profitability has been proposed [22].

To this aim, the productive characterization of this population is mandatory. However, there is a scarce resource allocation in the research of local genotypes [18], especially in turkeys, where there are few studies on egg quality in unconventional rearing [23]. For this reason, simple but cheap research approaches avoiding the breaking of the shell should be addressed. In this line, external quality attributes could act as an approach to overall egg quality. Egg weight and shape index are considered among the external parameters most closely associated with internal egg quality [23–25]. They are strongly correlated with albumen [25–29] and yolk quality [23,24,26,28–30], as well as other external attributes such as shell quality and porosity [27]. Additionally, literature has reported their association with overall hatchability [24] and their use for determining poult's sex [20]. Therefore, the study of egg weight and shape index must be proposed as a first approach to studying overall egg quality when scarce resources are available.

Hence, the present work aims to determine the influence of meteorological conditions and the lunar cycle on the external quality of eggs laid by outdoor Andalusian

turkey hens. This might enable the determination of the main environmental effects influencing egg quality in turkeys, which will lead to a deeper understanding of the interaction of the environment on alternative productions. This study is the pioneer in this field and will lead to further studies, as overall quality is approached from the study of egg weight and shape index. Additionally, the present work contributes to the productive characterization of the Andalusian turkey, which would contribute to the governmental recognition of the genetic resource as a breed.

## Materials and methods

### Study design

**Ethics approval.** The present work was performed according to the Declaration of Helsinki, the Directive 2010/63/EU of the European Parliament and of the Council of September 22, and the Spanish legislation (Royal Decree-national law 113/2013, of February 1). The experimental phase of the study eluded the scope of evaluation of the Ethics Review Committee of the University of Córdoba, as it did not comply with the legislation on animal protection in experimentation. The methods developed were under the ARRIVE guidelines and the authority of the Agropecuary Provincial Centre of Diputación de Córdoba (the Córdoba Provincial Council; Spain), where the study was conducted, granted permission.

**Flock housing and environmental conditions.** The experimental phase of the study was carried out in the facilities of the Agropecuary Provincial Centre of Diputación de Córdoba, in Andalusia (Spain) from February 2019 to April 2020. Eggs included in the study were laid by a flock of 22 mature turkey hens (aged 70 weeks of age at the beginning of the study) belonging to the conservation nuclei of the Andalusian turkey breed. Animals were not involved in further experiments but in the egg control production, and were kept in outdoor pens, simulating backyard conditions (Fig 1). Ad libitum water and commercial feed (15.20% crude protein, 4.10% calcium, 0.66% available phosphorus; metabolizable energy value=2420 kcal/kg) were provided to hens during the complete experimental phase.

**Work sample and quality measurements.** The sample comprised 194 eggs collected during the 14-month experimental phase. Egg collection and analysis was performed every 14 days. Therefore, eggs laid on 28 different days, representing an entire year, are included in the present analysis. Eggs were daily collected at the same time in the morning (08:00 a.m.) and quality analysis was then performed to ensure that measures were taken within the 24 hours after oviposition. Egg weight was measured using an electronic scale (Cobos, CSB-600C, Barcelona, Spain; precision: ± 0.01g.) The major diameter or long axis, considered as the maximum distance between the broad and narrow

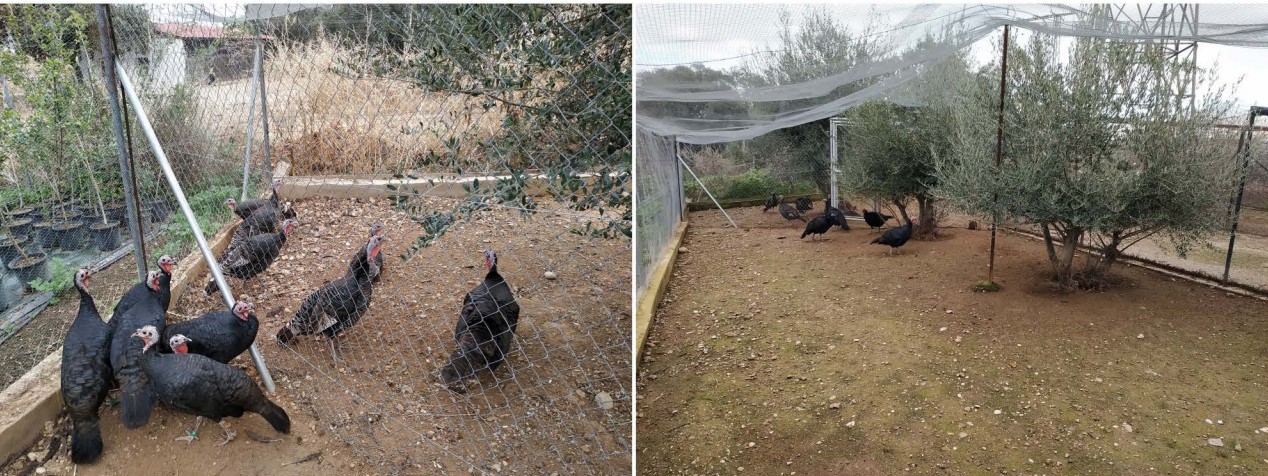

**Fig 1. Andalusian turkey hens (black and roan plumage) and outdoor pen in which the birds stayed during the study.**

poles, and the minor diameter or short axis, considered as the line of maximum width, were measured using a Vernier scale (Electro DH M 60.205, Barcelona, Spain). Diameters were employed to determine the shape index (SI) of the eggs, computed through the following formula:

$$SI = \frac{\text{Øm}}{\text{ØM}} \times 100,$$

where ØM is the major diameter and Øm is the minor diameter.

**Meteorological variables and moon phase.** Records were completed by collecting information on weather conditions and moon phases during the 24 hours before egg laying [31,32]. Information about the moon phase was obtained from the Astronomical Applications Department of the US Naval Observatory website (https://aa.usno.navy.mil/ accessed on 12 February 2024; Coordinates: 37°91′51″ N–04°70′91″ W). Moon phases were registered distinguishing seven categories: waxing gibbous, waning crescent, waxing crescent, waning gibbous, new moon, first quarter, and full moon. Moreover, the percentage of the moon-illuminated surface was recorded as well. Weather parameters were obtained from the official, historical data repository of the meteorological station located near the study site (https://datosclima.es/ accessed on 12 February 2024; Coordinates: 37°50′56″ N–04°50′48″ W). Parameters included were maximum temperature (°C), minimum temperature (°C), maximum pressure (mB), minimum pressure (mB), maximum gust speed (m/s), wind direction (°), average wind speed (m/s), photoperiod (h) and daily rainfall (l/m2).

## Statistical analysis of data

**Discriminant canonical analysis.** A discriminant canonical analysis (DCA) was performed to develop a statistical tool enabling the classification of eggs according to the influence of meteorological and moon phase events on external egg quality. To this aim, eggs were grouped according to their weight and shape index. Weight terciles were created from database observations and categories were identified as heavy eggs (100.06–85.77 g, tercile 1), medium eggs (85.69–77.33 g, tercile 2), and light eggs (77.31–61.25 g, tercile 3). For shape index (SI), eggs were classified attending to commercial standards developed in chicken as sharp (SI < 72), standard (72 < SI < 76), and rounded (SI > 76) [1,22]. Therefore, seven categories were obtained by combining egg weight and shape index: Sharp–1, Sharp–2, Sharp–3, Standard–1, Standard–2, Standard–3, and Rounded–3. No observations ascribed for the Rounded-1 and Rounded-2 categories were found in the database.

The egg weight and shape index combination were considered as the dependent variables in the DCA, while the aforementioned meteorological and moon phase parameters were included as explanatory variables. The canonical relationships among traits were graphically represented to exhibit group differences in a territorial map. Regularized forward stepwise multinomial logistic regression algorithms were used to perform the variable selection. Prior probability was computed to regularize priors according to the group sizes. All the analyses were run using the discriminant analysis routine of the Analyzing Data package of XLSTAT software (Addinsoft Pearson Edition 2014, Addinsoft, Paris, France).

**Preliminary multicollinearity testing.** Before the computation of discriminant canonical analysis, the multicollinearity assumption must be proved to detect redundancies. A redundant variable is one that explains the same source of variation as another already present in the study without contributing further additional variation. The presence of redundant variables in the analysis means the overinflation of variability in the analysis. Multicollinearity represents the linear intercorrelation between just two variables or between one linear variable and the linear combination of others [1]. In order to determine multicollinearity among variables, the variance inflation factor (VIF) is a parameter commonly employed in statistical analysis due to its ability to detect the increase in variance of a regression coefficient due to collinearity. The variance inflation factor (VIF) and tolerance have been employed in many studies in animal science research [1,18,22,32], following the formula:

$$VIF = \frac{1}{1 - R^2},$$

where R2 represents the determination coefficient of the regression equation. Values under 5 are recommended to avoid multicollinearity problems in variables [22].

**Discriminant canonical analysis efficiency and model reliability.** To define variables significantly contributing to the discriminant function, the Wilks' Lambda test was performed. As values tend to 0, the greatest contribution of a variable to the discriminant function. However, only significance ($\chi$2) values under 0.05 are considered to explain group adscription [1].

Due to the lack of equality in sample sizes, Pillai's trace criterion was selected for the assumption of equal covariance matrices in the discriminant function analysis [33]. Obtaining a significance equal to or under 0.05 is indicative of the statistical significance of the set of predictors [22].

**Variable dimensionality reduction.** A principal component analysis was employed before performing a DCA. This enabled the identification of those few most contributing variables in the effect of environment on egg weight and shape index.

**Canonical coefficients, loading interpretation, and special representation.** The percentage of allocation of each observation within its group was determined through the discriminant function analysis. Thus, according to the bibliography, only discriminant loading values ≥ |0.40| were considered to define substantive discriminating variables [34]. Variables exhibiting greater absolute coefficients within a set were those reporting a higher discriminant ability. Nonsignificant variables were removed before entering the function by the use of the stepwise procedure technique [35]. Subsequently, squared Mahalanobis distances and principal component analysis were performed using this formula:

$$D_{ij}^2 = \left( \bar{Y}_i - \bar{Y}_j \right) \; COV^{-1} \left( \bar{Y}_i - \bar{Y}_j \right),$$

where $D_{ij}^2$ represents the distance between population i and j; $COV^{-1}$ is the inverse of the covariance matrix of measured variable x, and both $\bar{Y}_i$ and $\bar{Y}_j$ represent the means of variable x in the ith and jth populations, respectively.

The Mahalanobis distance matrix obtained was then transformed into an Euclidean distance matrix, and a dendrogram representing the clusters within the environmental classification of egg shape and size was plotted. This was performed using the underweighted pair-group method arithmetic averages (UPGMA) from the Universität Rovira i Virgili (URV), Tarragona, Spain, and the Phylogeny procedure of MEGA X 10.0.5 (Institute of Molecular Evolutionary Genetics, The Pennsylvania State University, State College, PA, USA).

**Discriminant function cross-validation.** To test the reliability of the discriminant function, the leave-one-out cross-validation was performed. This cross-validation recorded the proportion of the right assignment of eggs within their clusters (weight and shape index categories) based on the environmental factors registered for each observation. The relative distance of each observation to the centroid of its closest cluster was employed. Press` Q statistic was performed to compare the discriminating power of the cross-validated function, following the formula:

$$Press'Q = \frac{(N - nK)^2}{N(K-1)},$$

where N is the number of observations in the sample, n represents the number of correctly classified observations and K is the number of groups. If Press' Q statistic value exceeds $\chi^2 = 6.63$, the cross-classification ability of the function is considered significantly better than chance.

## Results

### Preliminary multicollinearity test

Minimum pressure, maximum temperature and moon phase were found to be redundant variables (VIF > 5.0) and were discarded from further analysis. Parameters not showing multicollinearity problems are shown in Table 1.

**Table 1. Multicollinearity preliminary test of environmental factors influencing egg weight and shape index.**

| Statistics/Parameters | Tolerance (1 - R²) | VIF[a] |
|---|---|---|
| Maximum gust speed | 0.279 | 3.584 |
| Rainfall | 0.310 | 3.226 |
| Sunshine hours | 0.374 | 2.677 |
| Minimum temperature | 0.490 | 2.040 |
| Maximum pressure | 0.528 | 1.892 |
| % Moon illuminated | 0.722 | 1.385 |
| Wind direction | 0.820 | 1.220 |
| Average wind speed | 0.823 | 1.215 |

[a]Variance inflation factor (VIF) = 1 (not correlated); 1 < VIF < 5 (moderately correlated); VIF ≥ 5 (highly correlated).

### DCA model reliability

As shown in Table 2, the significance value of Pillai's trace criterion evidenced significant differences across egg categories, and thus, justified the implementation of the DCA.

### Canonical coefficients, loading interpretation, and spatial representation

Six canonical discriminating functions compose the DCA (Fig 2). The sum of the contributions of the first three functions accounts for 97.66% of the overall variance.

Fig 3 represents the loading value of each dependent variable (environmental factors) in the three first discriminant functions.

Table 3 represents the correlation matrix among climatic and lunar phase events on external egg quality and ranged from −0.537 to 0.574.

Results from the test of equality of group means are represented in Table 4. Variables were ranked attending to their discrimination ability, which increases with rising F values and decreasing Wilk's Lambda. Hence, variables not showing discriminating ability (p > 0.05) were rainfall, wind direction, maximum pressure, and the remaining moon phases (full moon, new moon, waxing crescent, waxing gibbous, and waning crescent). Moreover, the mean values of weather and moon phase variables for each egg quality-related group are shown in S1 Table.

Fig 4 represents the territorial map of observations once the values of environmental factors were replaced in the two first discriminating functions to obtain x and y-axis coordinates, for the first and second dimensions (F1 and F2), respectively. Subsequently, each observation was sorted and assigned across the different egg categories.

**Table 2. Results of the Pillai's trace test of equality of covariance matrices of the canonical discriminant functions allowing the implementation of the DCA.**

| Parameter | Value |
|---|---|
| Pillai's Trace Criterion | 0.523 |
| F (Observed value) | 2.208 |
| F (Critical value) | 1.369 |
| df1 | 48 |
| df2 | 1110 |
| Significance | <0.0001 |
| alpha | 0.05 |

F = Snedecor's F; df1 = numerator degrees of freedom for the F-approximation; df2 = denominator degrees of freedom for the F-approximation.

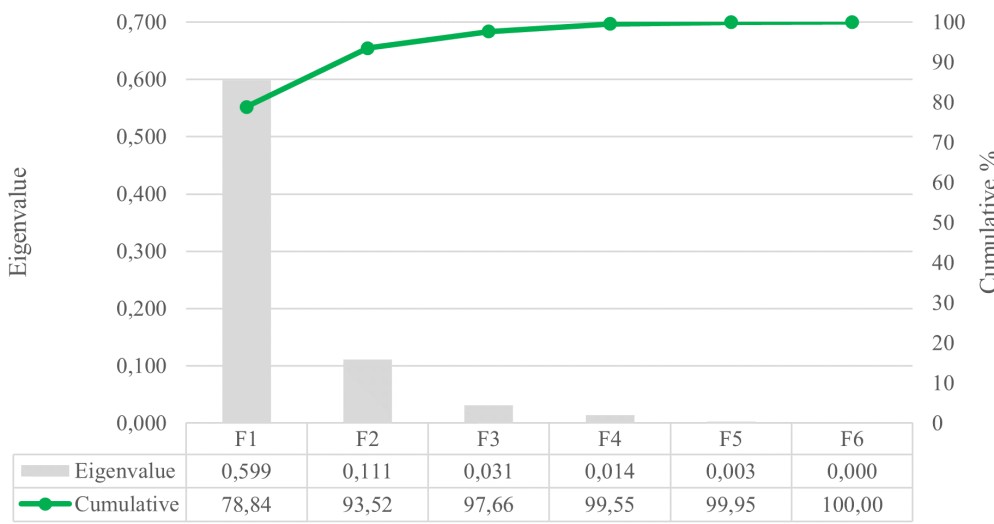

**Fig 2. Canonical variable functions and percentages of cumulative variance.**

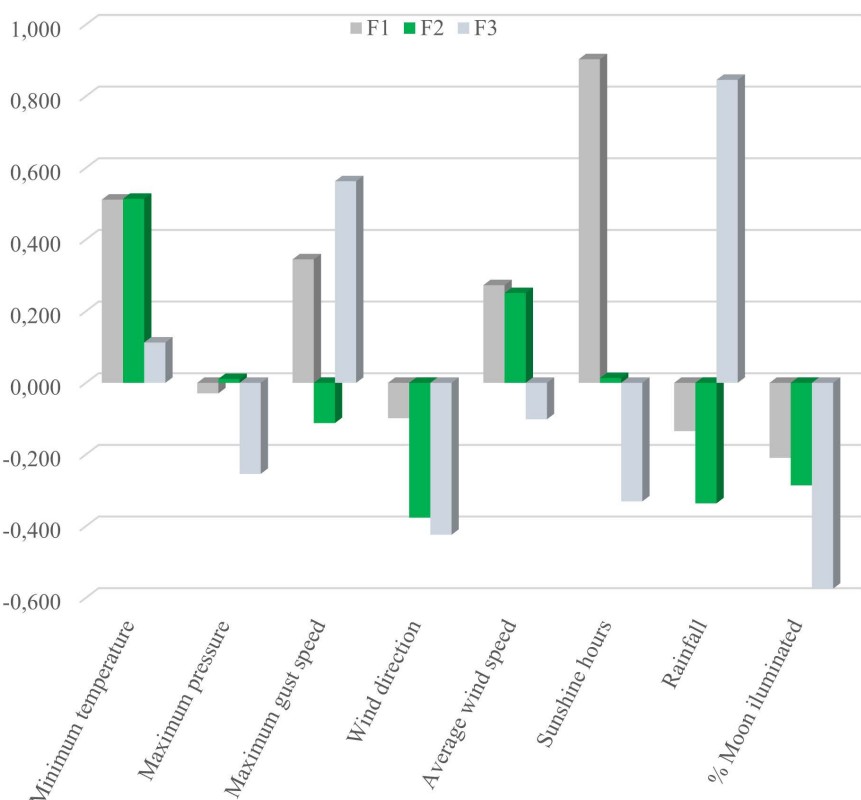

**Fig 3. Loading values of each environmental factor influencing egg shape in the discriminant canonical functions.**

**Table 3. Correlation matrix between climatic and lunar phase events included in the present study.**

| Variables | Minimum temperature | Maximum pressure | Maximum gust speed | Wind direction | Average wind speed | Sunshine hours | Rainfall | % Moon illuminated |
|---|---|---|---|---|---|---|---|---|
| Minimum temperature | 1 | −0.391 | 0.422 | 0.016 | 0.294 | 0.547 | −0.047 | −0.206 |
| Maximum pressure | −0.391 | 1 | −0.537 | 0.001 | −0.188 | 0.029 | −0.377 | 0.313 |
| Maximum gust speed | 0.422 | −0.537 | 1 | −0.215 | 0.303 | 0.181 | 0.574 | −0.080 |
| Wind direction | 0.016 | 0.001 | −0.215 | 1 | 0.081 | 0.150 | −0.257 | 0.197 |
| Average wind speed | 0.294 | −0.188 | 0.303 | 0.081 | 1 | 0.153 | 0.033 | 0.122 |
| Sunshine hours | 0.547 | 0.029 | 0.181 | 0.150 | 0.153 | 1 | −0.455 | −0.001 |
| Rainfall | −0.047 | −0.377 | 0.574 | −0.257 | 0.033 | −0.455 | 1 | −0.221 |
| % Moon illuminated | −0.206 | 0.313 | −0.080 | 0.197 | 0.122 | −0.001 | −0.221 | 1 |

**Table 4. Results for the tests of equality of group means to test for difference in the means across egg categories.**

| Test of Functions | Rank | Lambda | F | GL1 | GL2 | p-value |
|---|---|---|---|---|---|---|
| Sunshine hours | 1 | 0.691 | 13.950 | 6 | 187 | <0.0001 |
| Minimum temperature | 2 | 0.874 | 4.476 | 6 | 187 | 0.000 |
| Maximum gust speed | 3 | 0.944 | 1.841 | 6 | 187 | 0.043 |

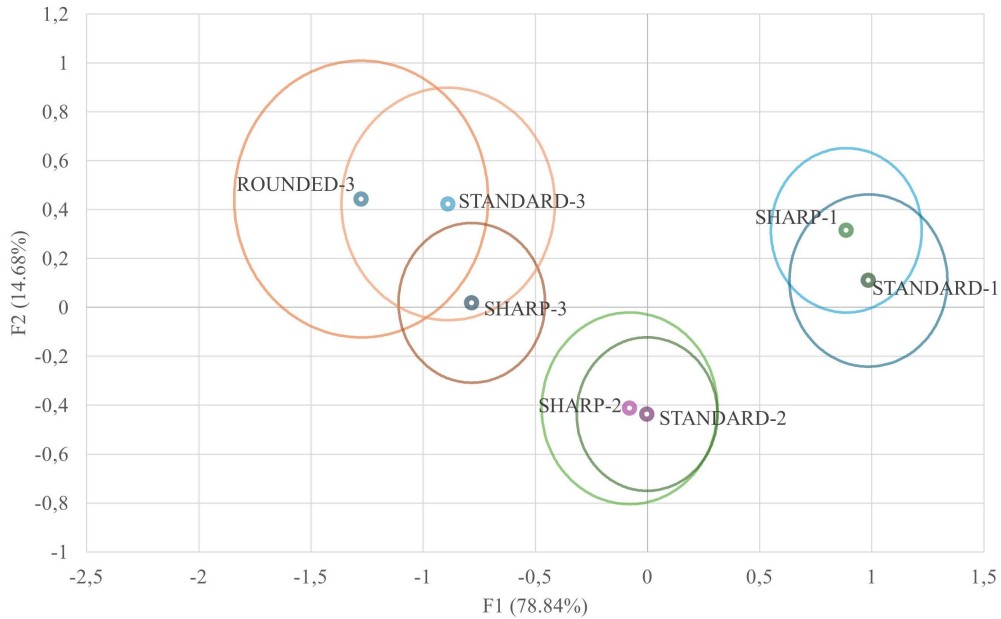

**Fig 4. Territorial map depicting the eggs considered in the canonical discriminant analysis sorted across egg weight and shape index categories.** Circles represent the centroids or canonical group means.

The Mahalanobis distances between categories were computed from the centroids generated by the discriminant routine. It is based on the fact that Mahalanobis distances can be estimated through the relative distance of a given observation to the centroid of its nearest group. Fig 5 represents the Mahalanobis distances transformed into squared Euclidean distances.

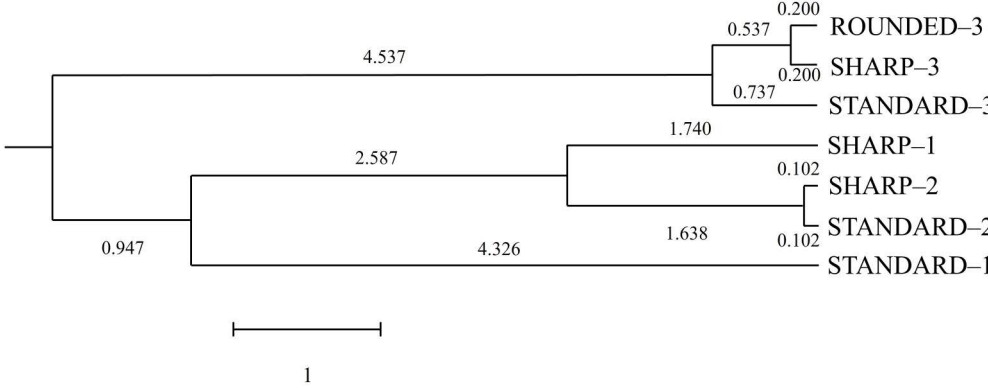

**Fig 5. Tree diagram based on Mahalanobis distances between environmental factors influencing egg weight and shape index.**

## Discriminant function cross-validation

Fig 6 reports the results obtained in the leave-one-out classification matrix. The model presented a 74.23% of correctly assignments of observations within their categories. Eggs belonging to the Rounded-3, Standard-1, Sharp-2, Sharp-1 and Standard-2 categories presented a high accuracy of correct matching (100.00, 93.55, 84.00, 79.41 and 74.36%, respectively). However, observations from the Standard–3 and Sharp–3 groups showed a moderately successful clustering rate (52.94 and 47.22%, respectively). A Press' Q value of 569.24 (N = 194; n = 144; K = 7) was obtained. Therefore, predictions were considered to be significantly better than those that would be obtained by chance at 95%.

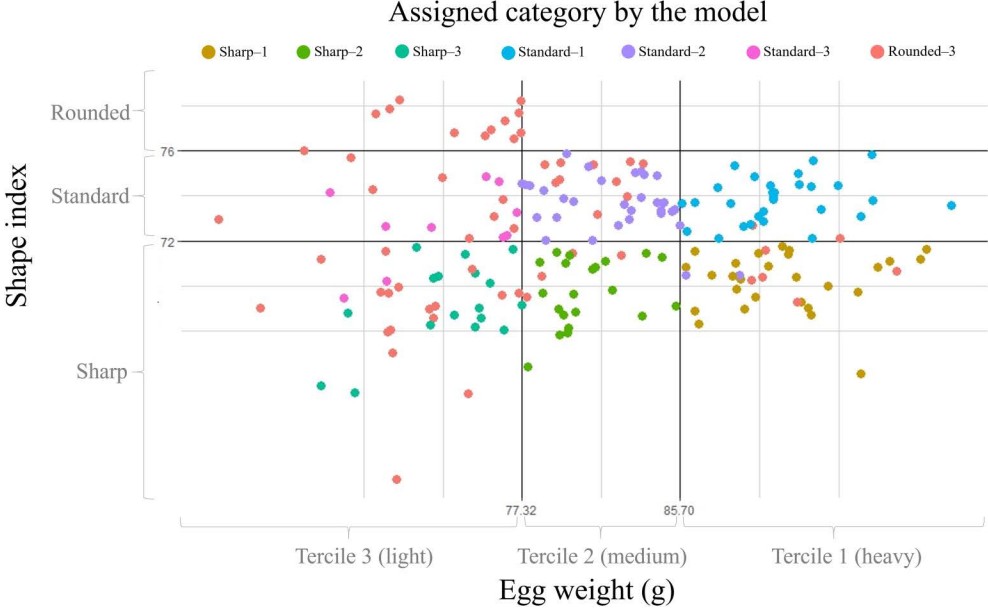

**Fig 6. Graphical representation of the proportion of correct assignments obtained using the classification and leave-one-out cross-validation matrices.**

## Discussion

The present work constitutes a pioneer study on several aspects. The effect of the environment on external egg quality in turkeys has been studied for the first time in this research. Moreover, this could be the first study of the species from a layer perspective in free-range systems, as well as one of the first studies addressing a local breed. The reason for this could be a consequence of the lack of resources allocated to the research of unconventional systems and indigenous genotypes, as well as the perception of turkey as a minor layer species in developed countries. Additionally, this work could provide a deeper understanding of the main environmental agents to which outdoor layers are exposed.

The preliminary test reported three variables exhibiting multicollinearity problems. Minimum atmospheric pressure was the first variable showing redundancy. In this respect, contradictory results are found in the bibliography. All pressure measures presented multicollinearity problems in a previous study [32], while maximum pressure (instead of minimum) was found to be redundant in another study [36]. Nevertheless, multicollinearity problems found in the present study could be a consequence of including other parameters to which pressure is highly correlated, such as wind speed [37], sunlight hours, and temperature [36]. The moon phase was the second variable exhibiting redundancy. In this case, due to the lack of previous reports, the inclusion of the moon's illuminated surface could be suggested as the source of multicollinearity. Finally, the maximum temperature was the last redundant variable. Regarding this aspect, no clear conclusions can be drawn from the literature. In the study developed by Iglesias Pastrana et al. [32], all the temperature parameters included exhibited redundancy while Marín Navas et al. [36] did not find multicollinearity among temperature variables. However, these authors reported that maximum temperature was highly correlated to minimum temperature, sunlight hours, and pressure, variables that are included in the present study.

Moderate positive and negative correlations among climatic and lunar variables are depicted in Table 3. The effect of maximum gust speed on egg quality is suggested to be generally associated with squall events. This variable was positively associated with rainfall (0.574), minimum temperature (0.422), and average wind speed (0.303), and negatively associated with maximum pressure (−0.537). The same trend seems to be described for rainfall, negatively associated with sunshine hours (−0.455), maximum pressure (−0.377), and wind direction (−0.257). Contrarily, the effect of minimum temperature seems to be associated with anticyclones and showed correlation with sunshine hours (0.547), average wind speed (0.294), and maximum pressure (−0.391). In this regard, similar correlations were found in a study of climatic variables influencing diversity parameters in a horse population [36]. Lastly, the illuminated surface of the moon correlated moderately to poorly with maximum pressure (0.313), wind direction (0.197), rainfall (−0.221), and minimum temperature (−0.206). Despite not showing discriminant capacity, interesting conclusions can be drawn, such as the possible evidence of the mediation of the lunar cycle in the influence of climatic variables and also on egg quality.

The test of equality of group means reported three variables to significantly perform as explanatory variables in the analysis. Sunshine hours was the variable reporting the highest discriminating power among egg groups. Similar results were described in a study on the correlations between meteorology and fowl egg quality, in which sunshine was the most influencing parameter [38]. Photoperiod is known to directly influence external egg quality [16], including egg weight, length, and width [38]. The general assumption is that, by increasing the photoperiod, higher yields are obtained, up to a certain limit [39]. Optimal light exposures to stimulate normal laying production in commercial turkey hens have been reported to be 14–16 hours, with negative effects occurring if 20 hours are exceeded [40]. This effect is firstly attributed to the increased general activity that photoperiod induces in animals, including feeding activity, which translates into a greater productive performance [41]. However, several hormonal mechanisms are underlying the effect of light on poultry performance. The hypothalamus-pituitary axis of birds is particularly sensitive to light, which directly stimulates GnRH hormone production [42]. This might explain the strong effect of photoperiod on sexual maturation, rate of egg production, and timing of the ovulatory cycle in hens [42]. Photoperiod also influences birds' circadian cycle [43] and about half of the melatonin of some bird species as the Japanese quail is produced in extra-pineal glands, such as the retinae [44]. Moreover, increasing daylengths enhances the release of follicle-stimulating hormone, which causes greater follicular growth

and development and stimulates ovulation [45]. Shell quality is also dependent on photoperiod, as light mediates mineral metabolism [46]. Long exposures to sunlight stimulate the synthesis of vitamin D, leading to a high accumulation of minerals in the shell [3]. This mechanism is even more accentuated in free-range systems, as direct sun exposures with natural UV radiation, alongside greater access to environment minerals, promote mineral metabolism leading to more deposition in the shell [3].

Attending to sunlight averages across egg categories, the present study reports that the heavier eggs (tercile 1) were obtained when the days were longer, while the opposite trend was observed for the lighter eggs. On the other hand, no clear effect of day length on shape index was described, but a slight positive tendency of elevated photoperiods producing standard-shaped eggs could be suggested within each weight group. In this manner, literature reports that photoperiod increases egg weight in turkey pullets during the first laying cycle [47]. However, no clear effect of day length on egg size has been described for adult turkey hens [40,47]. This lack of results in literature could be a consequence of the comparison of different artificial light programs that exceed natural day lengths. Therefore, these results might not be comparable to those obtained in the present study, in which animals were housed outdoors and only exposed to sunlight. This highlights the lack of studies involving turkeys in outdoor systems.

Finally, the high sensitivity to photoperiod obtained in the present study could suggest further conclusions. Indigenous genotypes are more adapted to harsh conditions and generally are more resistant to environmental fluctuations [2,3,18]. However, primitive genotypes preserve ancestral physiological mechanisms that were progressively eliminated through exhaustive selection, such as seasonality of production [48]. This is mainly ruled by the circadian cycle and pineal gland activity, which is severely atrophied in domestic animals compared to their wild ancestor [49]. Hence, the strong dependence on the photoperiod of this population could be evidence of the close relationship of the Andalusian turkey with the turkey's wild ancestor. For this same reason, the lack of discriminant ability of moon variables on egg quality is surprising. Since being a primitive breed with enhanced pineal activity kept outdoors, the influence of moonlight fluctuations on egg quality was expected [43,50]. This relationship has already been observed in some wild bird species, in which the moon cycle influences the laying rate, egg formation, and shell quality [51].

Minimum temperature was the second most discriminating variable in the analysis. This is in line with the extensive literature describing the effects of temperature on poultry egg quality [16,29,38]. In commercial turkey, the comfort zone ranges from 12 to 28 °C [52], and temperatures above or below end thermal equilibrium, leading to a state of chronic stress [53]. Despite research studies have focused primarily on heat stress, cold also plays an important role in egg production and quality, with a detrimental effect on egg weight, shell quality, and internal attributes [54]. Among the reasons underlying this effect is the decline in overall activity in animals, including feeding activity [54]. Cold stress also decreases feed efficiency [55], as energy is lost by allocating it to increased body temperature [56]. Moreover, cold stress generates a decrease in plasma insulin and an increase in corticosterone, hormones with opposite effects on energy metabolism [54,57]. This has a negative effect on anabolic metabolism, as insulin promotes glucose mobilization and tissue utilization, which benefits production-interest traits [57]. Additionally, cool decreases birds' ability to digest minerals [55] and reduces the intestinal absorption of some nutrients [54]. Additionally, the cold-stress response causes zinc deficiency [57], which mediates calcium storage.

However, variability in external egg characteristics could also be a consequence of the effect of cold on internal attributes, such as albumen [23]. Thus, a study developed in chicken species found that thermoneutral hens laid eggs that had greater albumen depositions, which occur in the equatorial region of the egg, increasing the egg's shape index (rounded eggs) and egg weight as well [58]. Moreover, eggs from cold-stressed hens are known to have a lower albumen height and yolk weight [54], possibly due to a reduced synthesis of both yolk and albumen proteins, vitellogenin and triglycerides, as well as a water composition [55]. This is consistent with the means by egg categories of the present study, as heavier eggs were obtained at warmer minimum temperatures. On the other hand, colder minimum temperatures produced both medium and small eggs, indifferently. In the same line, shape index quality seems to improve with warmer

minimum temperature, as they tend to produce standard eggs. This enhanced quality found when minimum temperatures are warmer suggests turkeys loose thermoneutrality in cold nights. This finding, together with the multicollinearity problems found in the maximum temperature variable, might prove that the Andalusian turkey population is more sensitive to cold than heat. This is evidence of the great adaptation of this breed to the extreme heat the Guadalquivir basin reaches [19,34], and warns breeders to focus on protecting during winter nights to avoid a decline in egg quality.

Maximum gust speed was the third and last variable reporting explanatory power in the analysis. Attending to the means by classes, there is a clear tendency for egg weight to increase with gust speed. Moreover, within weight groups, as gust speed increases, the egg-shape index declines. In this connection, the effect of wind on commercial layers has widely been studied [16,38]. However, no clear conclusions are drawn in the bibliography since both positive and negative effects are reported. A study reporting the benefits of air velocity on egg production suggested that wind acted as a heat stress alleviation effect [59]. On the other hand, another study found wind speed to reduce egg weight and decrease the shape index, which is consistent with present results [38]. Wind reduces locomotor and feeding activity [60], causes a negative water balance due to skin moisture losses [61], and predisposes to the onset of disease [62]. Nevertheless, in hot weather, wind acts as a relief factor and increases the laying rate [16,61]. However, studies were developed in indoor conditions, and no studies have been addressing the effect of wind on layers, even less in turkeys.

Finally, the leave-one-out cross-validation routine described an acceptable hit ratio, which seems to be more accurate as egg weight increases. Weather conditions producing light eggs categories (tercile 3) showed the worst results compared to the rest of the groups. Furthermore, the low inclusion of round eggs in the study (12 observations) and their presence in only one weight category (tercile 3), together with the high number of sharp eggs should be highlighted. This could be the consequence of the lack of a specific egg shape index standard for turkey species since the standards used worldwide for chicken were employed in the present work [34]. Thus, a possible deviation to the left of the shape indexes could be suggested in turkeys, at least in the present population, due to the consideration of a high percentage of eggs as undesirable, according to this standard.

## Conclusion

Results from the present work provide pioneer information about the effect of the environment on egg quality in free-range turkey eggs and contribute to the scarce literature approaching indigenous genotypes. On the other hand, the limitations of working with external attributes call for further studies including a comprehensive analysis of internal and external traits to better understand the influence of climate and the lunar cycle on egg quality. This paper demonstrates that alternative egg production faces different environmental constraints than inland egg production and affects them in different ways. Among the explicative parameters found in the DCA, sunlight exhibited the greatest discriminant ability. This supposes an interesting finding, as outdoor systems do not implement artificial photoperiod, and egg quality is suggested to have seasonal variations. Minimum temperature was the second most explanatory variable. The fact that lower temperatures worsen egg quality, together with the problems of multicollinearity of maximum temperature, show that this population is more sensitive to cold than to heat. This evidences the great adaptation of this breed to the Mediterranean basin and its potential for future climate challenges, as well as points farmers to minimize cold exposure to save egg quality. Maximum gust speed was the last discriminating variable, exerting an overall positive effect on egg weight and shape index. Although excessive air velocity worsens egg quality in commercial rearing, the wind seemed to exert a positive effect on this population and housing, possibly acting as a heat-relieving factor. Therefore, maximum egg quality seems to be achieved during spring and summer seasons, when long days with warm nights and wind acting a relieving factor are commonly found. Both maximum gust speed and minimum temperature were explanatory variables reporting a high correlation to others, which makes them particularly suitable for the study of weather's influence on egg quality. Finally, acceptable hit classification was obtained in the cross-validation test, except from the lighter egg categories. Additionally, different sample sizes may suggest the need to determine a specific egg shape index standard for the turkey, as those for chicken were employed due to the lack of turkey species.

## Supporting information

**S1 Table.  Mean values of weather attributes and moon phase for each egg category.**
(DOCX)

## Acknowledgments

This work would not have been possible without the assistance of ACPA (Asociación de Criadores del Pavo Andaluz), IFAPA (Instituto de Investigación y Formación Agraria y Pesquera), Diputación de Córdoba, and the PAIDI AGR 218 research group.

## Author contributions

**Conceptualization:** José Ignacio Salgado Pardo, Antonio González Ariza.

**Data curation:** José Ignacio Salgado Pardo, Antonio González Ariza.

**Formal analysis:** José Ignacio Salgado Pardo, Antonio González Ariza.

**Funding acquisition:** María Esperanza Camacho Vallejo.

**Investigation:** José Ignacio Salgado Pardo, Antonio González Ariza, José Manuel León Jurado.

**Methodology:** José Ignacio Salgado Pardo, Antonio González Ariza.

**Project administration:** Juan Vicente Delgado Bermejo, María Esperanza Camacho Vallejo.

**Resources:** Juan Vicente Delgado Bermejo, Antonio González Ariza, José Manuel León Jurado, María Esperanza Camacho Vallejo.

**Software:** José Ignacio Salgado Pardo, Antonio González Ariza, José Manuel León Jurado.

**Supervision:** Juan Vicente Delgado Bermejo, Antonio González Ariza, María Esperanza Camacho Vallejo.

**Validation:** José Ignacio Salgado Pardo, Antonio González Ariza.

**Visualization:** Juan Vicente Delgado Bermejo, Antonio González Ariza, María Esperanza Camacho Vallejo.

**Writing – original draft:** José Ignacio Salgado Pardo, Antonio González Ariza.

**Writing – review & editing:** José Ignacio Salgado Pardo, Juan Vicente Delgado Bermejo, Antonio González Ariza, José Manuel León Jurado, María Esperanza Camacho Vallejo.

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
