## [Decision Letter · Decision Letter 0]

Dear Dr. González Ariza,

Thank you for submitting your manuscript to PLOS ONE. After careful consideration, we feel that it has merit but does not fully meet PLOS ONE’s publication criteria as it currently stands. Therefore, we invite you to submit a revised version of the manuscript that addresses the points raised during the review process.

Dear Dr., Antonio González Ariza

Thank you for submitting your manuscript to PLOS ONE. After careful consideration, we have decided that your manuscript needs Major Revision.

Kind regards,

Prof. Lamiaa Mostafa Radwan, Ph.D.

Academic Editor

PLOS ONE

**Reviewer 1**

Kindly review the MS for a few mistakes/omissions according to my point of view, otherwise its okay. However this study is unique and leading to a new pathway to explore some of the myths related to lunar effect in scientific way. I congratulate the author/s.

**Reviewer 2**

1. The problem that prompted the research should be written at the beginning of the abstract before presenting the purpose of the study.

2. The introduction should be somewhat brief.

3. The age of the turkey flock used in the experiment to collect eggs should be mentioned.

4. Indicate the number of birds used in the experiment and the total number of the nucleus flock.

5. It is preferable to support research materials with images that illustrate each of the following: (the morphology of the turkey breed used, the frr range system followed in the study, and the grading of the seven egg groups).

6. The metabolize energy value represented in the commercial diet should be added.

7. Indicate in detail the number of days over which the eggs were collected.

8. Since the measurements were taken on the eggs 24 hours after oviposition, indicate whether they were stored properly. The storage temperature and relative humidity should be determined.

9- The number of cited references is excessive. It is preferable to reduce the number by excluding older references if there are more recent ones.

10- In Figure 1, the cumulative variance (%) is written on the vertical axis.

11- It is preferable to add the significance levels to the correlation values in Table 3.

12- In Figure 3, the graph illustrates what is meant by F1 and F2..

We look forward to receiving your revised manuscript.

Kind regards,

Lamiaa Mostafa Radwan, Ph.D.

Academic Editor

PLOS ONE

“This research was funded by FEDER project P20_00893 and, during the covering period of a pre-doctoral contract (FPU Fellowship, with code FPU21/00010), the Spanish Ministry of Science and Innovation.”

Additional Editor Comments:

Dear Dr., Antonio González Ariza

Thank you for submitting your manuscript to PLOS ONE. After careful consideration, we have decided that your manuscript needs Major Revision.

Kind regards,

Prof. Lamiaa Mostafa Radwan, Ph.D.

Academic Editor

PLOS ONE

Reviewer 1

Kindly review the MS for a few mistakes/omissions according to my point of view, otherwise its okay. However this study is unique and leading to a new pathway to explore some of the myths related to lunar effect in scientific way. I congratulate the author/s.

Reviewer 2

1. The problem that prompted the research should be written at the beginning of the abstract before presenting the purpose of the study.

2. The introduction should be somewhat brief.

3. The age of the turkey flock used in the experiment to collect eggs should be mentioned.

4. Indicate the number of birds used in the experiment and the total number of the nucleus flock.

5. It is preferable to support research materials with images that illustrate each of the following: (the morphology of the turkey breed used, the frr range system followed in the study, and the grading of the seven egg groups).

6. The metabolize energy value represented in the commercial diet should be added.

7. Indicate in detail the number of days over which the eggs were collected.

8. Since the measurements were taken on the eggs 24 hours after oviposition, indicate whether they were stored properly. The storage temperature and relative humidity should be determined.

9- The number of cited references is excessive. It is preferable to reduce the number by excluding older references if there are more recent ones.

10- In Figure 1, the cumulative variance (%) is written on the vertical axis.

11- It is preferable to add the significance levels to the correlation values in Table 3.

12- In Figure 3, the graph illustrates what is meant by F1 and F2.

Reviewers' comments:

Reviewer's Responses to Questions

**Comments to the Author**

1. Is the manuscript technically sound, and do the data support the conclusions?

Reviewer #1: Yes

Reviewer #2: Yes

2. Has the statistical analysis been performed appropriately and rigorously?

Reviewer #1: Yes

Reviewer #2: Yes

3. Have the authors made all data underlying the findings in their manuscript fully available?

Reviewer #1: Yes

Reviewer #2: Yes

4. Is the manuscript presented in an intelligible fashion and written in standard English?

Reviewer #1: Yes

Reviewer #2: Yes

Reviewer #1: Kindly review the MS for a few mistakes/omissions according to my point of view, otherwise its okay. However this study is unique and leading to a new pathway to explore some of the myths related to lunar effect in scientific way. I congratulate the author/s.

Reviewer #2: 1. The problem that prompted the research should be written at the beginning of the abstract before presenting the purpose of the study.

2. The introduction should be somewhat brief.

3. The age of the turkey flock used in the experiment to collect eggs should be mentioned.

4. Indicate the number of birds used in the experiment and the total number of the nucleus flock.

5. It is preferable to support research materials with images that illustrate each of the following: (the morphology of the turkey breed used, the frr range system followed in the study, and the grading of the seven egg groups).

6. The metabolize energy value represented in the commercial diet should be added.

7. Indicate in detail the number of days over which the eggs were collected.

8. Since the measurements were taken on the eggs 24 hours after oviposition, indicate whether they were stored properly. The storage temperature and relative humidity should be determined.

9- The number of cited references is excessive. It is preferable to reduce the number by excluding older references if there are more recent ones.

10- In Figure 1, the cumulative variance (%) is written on the vertical axis.

11- It is preferable to add the significance levels to the correlation values in Table 3.

12- In Figure 3, the graph illustrates what is meant by F1 and F2.

**Do you want your identity to be public for this peer review?** For information about this choice, including consent withdrawal, please see our Privacy Policy

Reviewer #1: **Yes: ** Dr Abdul Waheed

Reviewer #2: **Yes: ** Abdelmoniem Hanafy

---

## [Author Response · Author response to Decision Letter 1]

19 May 2025

Antonio González Ariza

Agropecuary Provincial Centre, Diputación Provincial de Córdoba

Department of Genetics, Faculty of Veterinary Sciences, University of Córdoba

Córdoba (Spain)

+34 679305661

angoarvet@outlook.es

16/10/2024

Dear Editor,

All the team responsible for this paper acknowledge the comments from the reviewers and editor, as they help to improve the quality of our manuscript. In the following paragraphs, we will describe and address how reviewers’ new recommendations were followed.

Reviewer 1

Kindly review the MS for a few mistakes/omissions according to my point of view, otherwise its okay. However this study is unique and leading to a new pathway to explore some of the myths related to lunar effect in scientific way. I congratulate the author/s.

We thank the reviewer for his/her kind comments.

Reviewer 2

1. The problem that prompted the research should be written at the beginning of the abstract before presenting the purpose of the study.

As requested, an introductory sentence highlighting the unawareness in the field has been added to the abstract to reinforce the aim of the study.

2. The introduction should be somewhat brief.

As requested, the introduction has been shortened from the initial 909 words to the current 684 words of length, including cites. Core concepts have remained while removing secondary information and simplifying sentences.

3. The age of the turkey flock used in the experiment to collect eggs should be mentioned.

The age of the hens that constituted the experimental flock has been added as requested.

4. Indicate the number of birds used in the experiment and the total number of the nucleus flock.

As requested, the number of the hens that constituted the experimental flock has been added.

5. It is preferable to support research materials with images that illustrate each of the following: (the morphology of the turkey breed used, the frr range system followed in the study, and the grading of the seven egg groups).

A new figure has been uploaded and included in the manuscript in which the appearance of the turkey landrace under study and the pen where animals were kept are visible. No additional pictures on the egg grading are included since we consider that the specific egg weights and shapes metrics differentiating categories are sufficiently described in the materials and methods.

6. The metabolize energy value represented in the commercial diet should be added.

As requested, the metabolizable energy value represented in the commercial diet has been added in the manuscript.

7. Indicate in detail the number of days over which the eggs were collected.

A deeper explanation of the collection and measurement protocol has been included in the material and methods, specifying the number of days of egg collection and analysis, as requested.

8. Since the measurements were taken on the eggs 24 hours after oviposition, indicate whether they were stored properly. The storage temperature and relative humidity should be determined.

Eggs were not stored between collection and quality measurement, they were directly analyzed after collection.

9- The number of cited references is excessive. It is preferable to reduce the number by excluding older references if there are more recent ones.

The number of references has been reduced from the initial 79 to the current 62 references.

10- In Figure 1, the cumulative variance (%) is written on the vertical axis.

The cumulative contribution to the variance explanation of the model is on the right vertical axis to describe their units (0 – 100%), different than those for the eigenvalues. The six discriminating functions generated in the model are represented on the horizontal axis.

11- It is preferable to add the significance levels to the correlation values in Table 3.

The correlation analysis was automatically generated as part of the discriminant routine and was included in the analysis as extra information about the interaction between the variables included. We suggest that, as a secondary approach in the study, we could avoid adding further detail to keep this table simple, since general conclusions can be extracted from the ‘r’ value.

12- In Figure 3, the graph illustrates what is meant by F1 and F2.

F1 and F2 are the first two (and more variance-explanatory) discriminating functions generated by the model. In this figure, coordinates were represented through their equations to represent the spatial representation of the group’s centroids. However, the variance explanatory potential (%) of each variable has been added in this figure.

To this end, we thank you for your time and attention and for considering this manuscript.

Should you require any further information do not hesitate to contact me via email or telephone at any time.

Yours faithfully,

Dr. Antonio González Ariza

---

## [Decision Letter · Decision Letter 1]

The effect of photoperiod, environmental temperature and wind speed on external quality of free-range turkey eggs

PONE-D-25-17693R1

Dear Dr. Antonio González Ariza,

We’re pleased to inform you that your manuscript has been judged scientifically suitable for publication and will be formally accepted for publication once it meets all outstanding technical requirements.

Kind regards,

Lamiaa Mostafa Radwan, Ph.D.

Academic Editor

PLOS ONE

Additional Editor Comments (optional):

Dear Dr., Antonio González Ariza

I am pleased to inform you that the manuscript has been accepted for publication.

Kind regards,

Prof. Lamiaa Mostafa Radwan, Ph.D.

Academic Editor

PLOS ONE

Reviewer 1

hope author/s have addressed all the queries/ suggestions as shown in the revised manuscript. Although more research is needed to quantify such facts, yet this would be pioneering work explaining the celestial impressions.

Reviewer 2

Thanks for quick response and doing the required comments, with my best wishes.

Most comments were mentioned, you complete it.

Reviewers' comments:

Reviewer's Responses to Questions

**Comments to the Author**

Reviewer #1: All comments have been addressed

Reviewer #2: All comments have been addressed

2. Is the manuscript technically sound, and do the data support the conclusions?

Reviewer #1: Yes

Reviewer #2: Yes

3. Has the statistical analysis been performed appropriately and rigorously?

Reviewer #1: Yes

Reviewer #2: Yes

4. Have the authors made all data underlying the findings in their manuscript fully available?

Reviewer #1: Yes

Reviewer #2: Yes

5. Is the manuscript presented in an intelligible fashion and written in standard English?

Reviewer #1: Yes

Reviewer #2: Yes

Reviewer #1: I hope author/s have addressed all the queries/ suggestions as shown in the revised manuscript. Although more research is needed to quantify such facts, yet this would be pioneering work explaining the celestial impressions.

Reviewer #2: Thanks for quick response and doing the required comments, with my best wishes.

Most comments were mentioned, you complete it.

**Do you want your identity to be public for this peer review?** For information about this choice, including consent withdrawal, please see our Privacy Policy

Reviewer #1: **Yes: ** Dr. Abdul Waheed

Reviewer #2: **Yes: ** Dr. Abdelmoniem Hanafy

---

## [Editor Report · Acceptance letter]

PONE-D-25-17693R1

PLOS ONE

Dear Dr. González Ariza,

I'm pleased to inform you that your manuscript has been deemed suitable for publication in PLOS ONE. Congratulations! Your manuscript is now being handed over to our production team.

Kind regards,

on behalf of

Prof. Dr. Lamiaa Mostafa Radwan

Academic Editor

PLOS ONE